# Poor Sleep Quality Experience and Self-Management Strategies in Fibromyalgia: A Qualitative Metasynthesis

**DOI:** 10.3390/jcm9124000

**Published:** 2020-12-10

**Authors:** Carolina Climent-Sanz, Genís Morera-Amenós, Filip Bellon, Roland Pastells-Peiró, Joan Blanco-Blanco, Fran Valenzuela-Pascual, Montserrat Gea-Sánchez

**Affiliations:** 1Department of Nursing and Physiotherapy, University of Lleida, 25198 Lleida, Spain; carol.climent@udl.cat (C.C.-S.); genis.morera11@gmail.com (G.M.-A.); roland.pastells@udl.cat (R.P.-P.); joan.blanco@udl.cat (J.B.-B.); fran.valenzuela@udl.cat (F.V.-P.); montse.gea@udl.cat (M.G.-S.); 2Grup de Recerca de Cures en Salut, GRECS, Institut de Recerca Biomèdica de Lleida, Fundació Dr. Pifarré, 25198 Lleida, Spain; 3Grup d’Estudis Societat, Salut, Educació i Cultura, GESEC, Department of Nursing and Physiotherapy, University of Lleida, 25198 Lleida, Spain

**Keywords:** fibromyalgia, Symptom Management Theory, symptom experience, poor sleep quality, metasynthesis, qualitative research

## Abstract

Poor sleep quality is a major concern and a highly prevalent symptom in fibromyalgia. We aimed to develop a metasynthesis of qualitative studies to assess how people diagnosed with fibromyalgia experience and manage poor sleep quality following the concepts of the Symptom Management Theory. The principles of metasynthesis established by Sandelowski and Barroso were utilized. A pre-planned comprehensive search was implemented in PubMed, Scopus, ISI WebofScience, and Cinahl Plus databases. The methodological quality was assessed following the CASP Qualitative Checklist. The findings of the studies were subjected to a metasummary and a metasynthesis. Seventeen studies were included in the metasynthesis. Two overarching themes were pre-established: (1) experience of poor sleep quality in Fibromyalgia and (2) poor sleep quality management strategies in Fibromyalgia. Four sub-themes emerged from the results: (1) evaluation of poor sleep quality, (2) response to poor sleep quality, (3) management strategies to favor sleep, and (4) managing the consequences of a sleepless night. Poor sleep quality is a severe and disabling symptom that negatively impacts the general health status of people diagnosed with FM. Prescribed treatments are commonly seen as ineffective and self-management strategies are a last resort and do not show beneficial effects.

## 1. Introduction

Fibromyalgia (FM) is a chronic health condition mainly characterized by chronic pain localized in multiple body areas [1]. The prevalence of FM in Spain is estimated at around 2.4% among the adult population over 20 years of age [2]. In terms of gender, FM is mainly diagnosed in women, representing 80–90% of the cases [3]. Moreover, studies comparing the prevalence of FM in rural and urban areas showed that people living in rural areas are more likely to be diagnosed with FM [4,5,6,7,8,9].

Although the etiology of FM is still an issue of scientific debate, it is classified as a central sensitivity syndrome (CSS) [10,11] given the evidence of a hyperexcitable central nervous system that alters the processing of nociceptive stimuli and the modulation of pain [12,13]. Furthermore, central sensitization may be contributing to the development and maintenance of the multiple medically unexplained symptoms associated with FM and that have a great impact on the psychosocial functioning of people suffering from it [13,14]. 

Other symptoms such as fatigue, poor sleep quality, and cognitive complaints are also highly prevalent among people diagnosed with FM [1,15,16]. Specifically, sleep disturbances are conceived as troublesome [17] and prevalence studies show that only 11.2% of people diagnosed with FM report having good sleep quality [18,19] while between 65 and 99% manifest problems in initiating and/or maintaining sleep [19,20].

Poor sleep quality has a profound impact on the experience of pain by increasing pain intensity [21] and pain cognitive-affective processes such as pain catastrophizing [22]. Poor sleep quality also impacts negatively on fatigue, cognitive problems [23], quality of life, and social functionality in people with FM [24]. A previous qualitative study showed that people diagnosed with FM perceived that poor sleep quality had a great impact on work performance because of their increased need for diurnal rest. Moreover, the increased need for diurnal rest also prevents these patients from engaging in social activities [25].

Currently, there are no curative treatments for FM and the available approaches are focused on symptom management [26]. Although it seems that multi-component treatments offer greater benefits, their effectiveness varies among patients. Additionally, as no cure is available, patients must reinterpret their roles in the management of FM and understand that the development of self-management strategies is essential to improve their general health status [26]. In accordance with the “European League Against Rheumatology” revised recommendations for the management of FM [27], educational interventions are essential for the initial management of this syndrome. However, the available pain management programs based on patient education show limited coverage of sleep education [28], emphasizing the need for developing new educational approaches integrating the topic of sleep. 

In health conditions such as FM, in which poor sleep quality shows a strong tendency towards chronification [29], exploring the cognitive and behavioral factors that act as perpetuators of such symptoms could be essential to develop effective management strategies. Moreover, understanding how people diagnosed with FM experience and manage poor sleep quality could help health providers to contextualize and conceptualize the content of the sleep educational programs to this health condition. 

Qualitative research provides an opportunity to engage the participation of people diagnosed with FM in the development of future treatment approaches, especially those based on educational interventions aimed at changing cognitive and behavioral factors associated with the experience of such symptoms. 

Qualitative studies are rarely aimed at having a direct impact on healthcare practice or policymaking. However, conducting systematized reviews of qualitative studies could be a valuable method in facilitating the transferability of qualitative data to improve healthcare attention [30]. 

Although the vast majority of meta-syntheses of qualitative studies are published in the fields of nursing and sociology, there is a clear trend towards the publication of metasynthesis in all disciplines of health science [31].

Therefore, the authors considered it appropriate to carry out a metasynthesis aimed at summarizing the available qualitative research exploring the experience and management of poor sleep quality in people diagnosed with FM. Carrying out this metasynthesis could improve the understanding of the phenomenon of poor sleep quality in the context of FM and provide valuable information for the development of treatment strategies. 

## 2. Methods

The metasynthesis was reported according to the “Enhancing Transparency in Reporting the Synthesis of Qualitative Research” (ENTREQ) statement recommendations [32].

### 2.1. Aim

We aimed to develop a metasynthesis of qualitative studies to evaluate how people diagnosed with FM experience and manage poor sleep quality.

### 2.2. Methodological Approach

In accordance with Sandelowski and Barroso [33], we integrated the findings of original qualitative research reports through a metasynthesis approach, including both a metasummary and a metasynthesis of the findings. A qualitative metasynthesis must ensure an interpretative integration of the findings in a way that “the integrations are more than the sum of parts in that they offer novel interpretation of findings that are the result of interpretive transformations” [34]. The six steps of a qualitative metasynthesis established by Sandelowski and Barroso [33] were followed: (1) formulating the review question, (2) conducting a systematic literature search, (3) screening and selecting appropriate research articles, (4) analyzing and synthesizing qualitative findings, (5) maintaining quality control, and (6) presenting findings.

### 2.3. Research Question

There is a necessity for increasing the understanding of the phenomenon of poor sleep quality in the context of FM. Therefore, this metasynthesis was developed to answer the research question, “How do people diagnosed with fibromyalgia experience and manage poor sleep quality?”

### 2.4. Approach to Searching

We implemented a pre-planned approach to searching, developing a comprehensive search strategy for gathering the available qualitative reports about the experience and management of poor sleep quality in people diagnosed with FM.

### 2.5. Inclusion Criteria

For inclusion in this metasynthesis, the qualitative reports were required to fulfill the following criteria: (1) qualitative or mixed methods research in which a qualitative phase was carried out including adult people diagnosed with FM, (2) studies totally or partially exploring the experience and/or management related to poor sleep quality in adult people diagnosed with FM, (3) studies published in English or Spanish since 1990.

### 2.6. Data Sources

Systematized searches were performed on the computerized databases PubMed, Scopus, ISI WebofScience, and Cinahl Plus. The reference lists of the included studies were scanned to identify additional studies. 

The last update of the search was carried out on 3 January 2020.

### 2.7. Electronic Search Strategy

The “Peer Review of Electronic Search Strategies” guideline recommendations were followed to develop the search strategy [35]. We combined MeSH (and their equivalent in other databases) and free-text terms such as “fibromyalgia”, “sleep”, “sleep quality”, “qualitative research”. The complete search strategy for PubMed is presented in Appendix B.

### 2.8. Study Screening Methods and Appraisal of the Methodological Quality

The study screening process and the appraisal of the methodological quality were carried out through a peer-review process in which two authors (CCS and GMA) screened the titles and abstracts of the retrieved reports in the first phase and the full-text in the second phase. After the study selection process, the authors appraised the methodological quality using the Critical Appraisal Skills Program (CASP) Qualitative Checklist [36]. The entire process was supervised by a third author (MGS) to resolve discrepancies.

### 2.9. Analyzing and Synthesizing Qualitative Findings

The synthesizing process was implemented considering two approaches proposed by Sandelowski and Barroso [33]: the qualitative metasummary, defined as “a quantitatively oriented aggregation of qualitative findings”, and the qualitative metasynthesis, that is “an interpretive integration of qualitative findings that are themselves interpretive syntheses of data” [33].

For the qualitative metasummary, we reported the characteristics of the included studies, the frequency of appearance of each of the target findings, the interstudy frequency effect sizes (i.e., representation of sub-themes in individual studies), and the intrastudy intensity effect sizes (i.e., individual studies’ contributions to sub-themes).

To metasynthesize the findings, the quotations of adult people diagnosed with FM were established as the target finding and were imported, structured, and analyzed using the qualitative analysis software Nvivo12 Plus (QRS International, Melbourne, Australia).

In this metasynthesis, we took the Symptom Management Theory (SMT) [37] as a biopsychosocial conceptual framework. The SMT is a deductive, middle-range theory developed by Larson et al. [38] providing a new model for nurses to assess the patients’ symptom experience in a comprehensive and multidimensional manner. Moreover, the SMT is also aimed at providing a framework for the development of symptom management strategies and to evaluate their outcomes in the biological, psychological, and social spheres of the patient. 

As previously stated, FM is a complex chronic health condition for which curative treatments are currently nonexistent. Therefore, the main approach is focused on developing symptom management strategies that help to alleviate the severity of symptoms such as pain, poor sleep quality, fatigue, and mood disturbances. Because of the latter, a symptom-focused conceptual framework could help us to understand how people diagnosed with FM experience the symptoms that characterize this health condition.

## 3. Results

The search in the different databases produced a total of 541 results. After the removal of 112 duplicates, 429 reports were screened against title and abstract and 404 were excluded. Finally, 25 reports were assessed for full-text eligibility. Of these, 14 complied with the inclusion criteria and were included in the analysis. Additionally, three reports were retrieved from the reference lists of the included studies. Therefore, 17 studies were finally included in the present metasynthesis. The results from the processes of identification, screening, eligibility, and inclusion of studies are presented in Figure 1 following the Prisma statement [39].

### 3.1. Metasummary

The characteristics of the included reports, the methodological quality, and the intrastudy intensity effect sizes, and interstudy frequency effect sizes of sub-themes are presented in Table 1 and Table 2, respectively.

### 3.2. Metasynthesis

The experience of poor sleep quality as a symptom of FM is presented into two pre-established overarching themes responding to the components of the “symptom experience” and “symptom management strategies” domains of the Symptom Management Theory. Therefore, the results were organized into the following overarching themes: (1) experience of poor sleep quality in FM and (2) poor sleep quality management strategies in FM (complete list of themes, sub-themes, codes, and quotations is presented in Appendix A.

#### 3.2.1. Experience of Poor Sleep Quality in FM

According to the SMT, symptom experience is determined by three major factors: symptom perception, evaluation, and response [37,38,56]. The perception, evaluation, and response to a symptom depend on the characteristics of each person, the physical, social, and cultural environment, as well as their health and illness status. Likewise, it is important to highlight that these three processes show a bidirectional relationship as all of them can influence or be influenced by the others and do not necessarily have to follow a predictable pattern of behavior [37,56].

This overarching theme was organized into two sub-themes revolving around two of the major factors that determine the symptom experience as described in the SMT: (1) evaluation of poor sleep quality and (2) response to poor sleep quality.

(1)Evaluation of Poor Sleep Quality

Based on the SMT, the evaluation of a symptom is understood as the meaning that a person attributes to the experienced symptom in terms of effects, severity, temporality, cause, and treatability [37,56].

*Poor sleep quality is a severe symptom of FM.* People diagnosed with FM sometimes feel that poor sleep quality is the worst symptom of their health condition [17,53,54], and it is associated with negative cognitive responses such as frustration and hatred towards life [53]. Being unable to sleep properly is perceived as a betrayal of the body which may indicate that, at least in some cases, people suffering from FM consider that they have no control over the symptom [54]. In other cases, participants reported that “sleep is such a gift” and that they “never take it for granted”, reflecting that sleep problems are particularly severe in the context of FM.


*“Sleep, or lack of it, is the worst thing about this condition for me […] It’s just another way my body has betrayed me.”*


According to the reports of the participants, the most common sleep disturbances associated with FM are those related to the maintenance of sleep, which is also associated with the feeling of being continuously sleep-deprived [17,43,53]. While participants were arguing that their sleep problem is mainly related to sleep maintenance, others reported problems falling asleep [17]. Moreover, the participants described problems such as having spams in the lower extremities that may be indicative of an underlying sleep disorder [17].


*“I can, initially, go to sleep, but it’s staying asleep that’s very hard. And then, I got to sleep tired and I wake up exhausted, and it’s frustrating.”*


*Perceived effects of poor sleep quality in other symptoms of FM.* People diagnosed with FM commonly identify poor sleep quality as one of the symptoms that has the greatest impact on fatigue, pain, cognitive functioning, ability to manage symptoms, eating behavior, and symptom flare-ups.

In general, people diagnosed with FM experience fatigue more acutely after a night of poor sleep quality, making them feel intense sleep pressure. However, they must wake up in the morning despite feeling tired to fulfill their working schedules, and, consequently, fatigue ends up impacting their functional capacity in the workplace [40,48].


*“Sleeplessness nights ... can you see the black circles under my eyes, because I wake up at 2 o’clock in the morning, then I wake up and feel right awake...and the next day I feel tired, I just want to sleep on but I have to get up and go to work ... I am extremely physically tired ... I am so tired I just can’t function as a normal person.”*


Participants reported that when they are very tired, they experience pain and small muscle cramps, being unable to fall asleep [53]. In some cases, the participants considered that the only solution to reducing fatigue levels is having a good rest at night [46], whereas, in others, the participants indicated that fatigue is maintained or appears even when it is possible to have a satisfactory amount of sleep [43,46,50,51,52].


*“The fatigue, it’s number one, because I can deal with the pain, at least up to a certain point, but the fatigue there’s nothing you can do besides sleep. There is no way to help that. There’s no pill you can take, there’s no medicine.”*



*“I could sleep 20 hours and still be tired. That is terrible.”*


Regarding the sleep–pain relationship, the general belief was that there is a bidirectional relationship between poor sleep quality and pain, so that one night of not having sufficient sleep or good sleep quality results in an increase in the intensity of pain the next day, which, at the same time, prevents a good rest at night [17,47]. One of the participants hypothesized that poor sleep quality affects her joints and muscles so that they do not “rest” properly, increasing pain intensity [40].


*“I feel constantly in pain, which obviously when I don’t get enough sleep will aggravate that, and then because I’ve aggravated pain I don’t get enough sleep. So I am on a vicious cycle, I can’t sleep properly because of the pain, and I can’t, because I am not sleeping, I then get in more pain.”*


Participants sometimes establish a direct connection between pain experience and poor sleep quality [53]. However, it is also argued that experiencing pain prevents them from finding a resting position that facilitates sleep [43,53]. Related to the latter, participants were concerned that the inability to find a comfortable sleep position results in a constant movement that ends up impacting the quality of the sleep of their bedfellows. Consequently, it makes them barely able to share the bed with their partner [53].


*“I don’t hardly ever sleep with my husband anymore, because I disturb his sleep so much of the time with my tossing and turning, trying to get comfortable, getting in and out of bed, because I can’t get comfortable.”*


Regarding the poor sleep–pain–fatigue cluster, it was described as a vicious circle in which insufficient sleep results in an increase in pain intensity the next day, which, at the same time, leads to a state of fatigue that prevents a good rest at night [51]. In other cases, it was reported that not moving during the night is the main cause of increases in pain intensity. Thus, staying awake favors continuous movement that seems to decrease pain although, in consequence, increases fatigue [50].


*“It’s not just the pain and the fatigue...it’s the nonrestorative sleep...it’s a vicious circle because if you don’t get enough sleep you feel pain more acutely...you’re more tired and unable to sleep well.”*


In addition to fatigue and pain, people diagnosed with FM believed that poor sleep quality also affects cognitive functioning [40,47] and symptom management abilities [17]. Likewise, poor sleep quality alters eating behaviors [40] and causes symptom flare-ups [42].


*“I will be thinking and, and trying to explain stuff to you, but my mind will just go completely blank. That gets worse on certain days, obviously with less sleep, but on other days I can sort of string together.”*


*Beliefs about the temporality and cause of poor sleep quality.* The perception of people diagnosed with FM is that the sleep problems appeared simultaneously with other symptoms of FM [17]. Regarding the cause of poor sleep quality, different beliefs were identified. While there were participants unable to identify a cause for their sleep problems [17], others pointed out that working night shifts during a long period resulted in problems initiating and maintaining sleep [45].


*“[After a night shift] I slept maybe an hour or hour and a half a day … you know how it is when you don’t sleep practically at all for weeks and months … In fact, I was practically sleepless for years …”*


*Meaning of good sleep quality.* For people diagnosed with FM, good sleep quality is mainly related to feeling renewed upon waking and having the energy to face their daily tasks [40]. Another important aspect that allows them to identify good sleep quality is the feeling of "disconnection" with the environment during the night or not remembering waking up due to noise [40].


*"It’s that sensation of really I have switched off, I am not aware of anything. That you know, those three hours where maybe the following day my husband said to me, “Oh did you hear the thunderstorm last night?” “No,” because it happened on those three hours and I didn’t hear anything. I didn’t hear the thunderstorm, I didn’t notice the light, nothing, and that is for me a proper sleep. When I’m aware of everything else I’m not, and I get up noticing that I have not slept properly.”*


(2)Response to Poor Sleep Quality

The SMT describes the symptom response as a factor associated with the person’s reactions to a symptom at the physiological, psychological, and social levels [37,56]. In this metasynthesis, and in accordance with the results, only the psychosocial factors were explored and described. 

*Feeling frustrated and like a failure.* People diagnosed with FM consider that constantly waking up is the worst thing about their health condition [17,53] and usually develop feelings of frustration and despair. Participants argue that sleep problems affect their functional capacity in the workplace, which causes them anxiety and a feeling of constant failure [17]. Other factors that lead to a feeling of frustration are not getting enough sleep [17], the inability to fall asleep, and extreme daytime fatigue due to poor sleep quality [43]. In some cases, participants even express hatred towards their health condition, the inability to satisfy their sleep needs being a central factor contributing to this feeling [53].


*“The not sleeping and then not being able to function the next day when you need to perform at work …—when you’re being paid and you’re meant to work and you can’t function, it’s horrible, it’s really horrible because you feel like a failure.”*


Participants believe that lack of sleep is aging them, which leads to feelings of hatred towards their life [41]. Moreover, they believe that there is no possible solution for their sleep problems [54] and that the impact of poor sleep quality on their functionality is an “absolute nightmare” [41].


*“I don’t feel like I can sleep. ... This is aging me, I can feel it. Sometimes I just hate life.”*


*Fear of going to bed.* According to the participants’ reports, people diagnosed with FM can develop a fear of not sleeping. This factor has a direct impact on their personal life, leading them to consider motherhood for that fear of not being able to meet their sleep needs [17]. The participants also develop a fear of the bedroom and fear of going to bed [17,54], reflecting the great impact of poor sleep quality at the emotional level.


*“Sleep, or lack of it, is the worst thing about this condition for me. I have christened my bedroom “the torture chamber.”*


Concerns about poor sleep quality were reported to manifest constantly, generating ruminating thoughts and fear of going to bed [17].


*“If there’s something on my mind, that makes me a bit worried about sleeping, I don’t know what it is but I sort of need to try and get to the bottom of it, but I sort of have this fear of going to bed.”*


#### 3.2.2. Poor Sleep Quality Management Strategies in FM

Developing symptom management strategies is not only about preventing, diminishing, or delaying the occurrence of negative outcomes, but it is also about controlling and influencing the whole symptom experience [37,56].

This overarching theme was organized into two sub-themes: (1) management strategies to favor sleep and (2) managing the consequences of a sleepless night.

(1)Management Strategies to Favor Sleep

This sub-theme describes the strategies that people diagnosed with FM employ to initiate and maintain sleep during the night. 

*Medication: from dependency to rejection.* Taking medication is one of the most frequent strategies among people diagnosed with FM to fall asleep and some of the participants report that not taking (sleep) medication prevents them from falling asleep [17,50,53]. While drug dependency behaviors can be intuited [17,49], participants also reported that they resort to taking medications occasionally when they feel they need them [48]. However, it also seems that there are attitudes of rejection towards sleeping medications, mainly due to side effects [45] and to the low effectiveness favoring good sleep quality [17]. 


*“I am on Amytriptiline to help with my sleep and other tablets … I only take them when it is right for my body. I don’t like to be taken as a guinea pig. I don’t trust staff to deal with me in that way. I am the only person who knows what it feels like to be ill, and what is good for my body, I don’t like other people to tell me and to control me.”*


*Self-management: behavioral adaptations.* A common strategy that favors sleep, and that participants recognize as essential, is to establish regular sleep schedules [17,55]. However, in some cases, having to constantly monitor the organization of sleep schedules and their activities of daily living (ADLs) generates anxiety [17].


*“I know that it’s an absolute crucial thing to get a good night’s sleep; to go to bed at the same time; eating on time... so everything has to be regular. And the more regular it is, the better I feel.”*


Participants also adopt other strategies such as the use of earplugs [17] or relaxation techniques [54]. However, these strategies are perceived as ineffective and, sometimes, unjustified. 


*“I put earplugs in and I find when I put earplugs in I seem to sleep different … although I don’t like things in my ears, if it means I’m going to sleep a bit better, so whether that’s about noise, although it’s very quiet where I live, so I don’t know, I think I’m just looking for answers.”*


In dealing with awakenings during the night, two opposite strategies were identified [17]. While participants reported that they get out of bed to avoid developing attitudes of rejection towards the bed, others stay in bed and try to fall asleep again. Regarding the latter strategy, it is usually perceived as ineffective and the participants end up turning on the television to try to refocus their attention. 


*“Sometimes I think I can get back to sleep, so I wait to see if I can and I think come on and I just lie there and hopefully I can get back to sleep easily again, um, but more often than not, I can’t so after I try for about 15 to 20 minutes and if I can’t get back to sleep after that time then as I say I put the television on and it’ll refocus me and if I fall back to sleep, good.”*


#### 3.2.3. Managing the Consequences of a Sleepless Night

Managing the consequences of a sleepless night described how people diagnosed with FM cope with the impact of poor sleep quality during the daytime.

*Medication: finding the balance between benefits and side effects.* The participants usually reported relying on drugs in the hope that they will help them to “continue to have a life” even if they cause side effects [51]. In other cases, it was commented that taking analgesics helped them to fall asleep during the day, providing a boost of energy to face ADLs during the afternoon [52].


*“I try to get out of bed every day regardless of how I feel. I’m trying to function, so I take the various medications … in the hopes that while it causes other problems it will at least allow me to continue to have a life.”*


*Resting and relaxing during the day.* The most generalized strategy is daytime rest, which helps in decreasing fatigue and allows them to cope with evening tasks [17,50,53,57]. However, it was also reported that daytime rest could be more a necessity than a strategy due to the extreme fatigue experienced after a sleepless night [52]. Moreover, the use of lavender scent while taking a hot bath in the morning helps to reduce morning stiffness and better cope with the day [44].


*“I might have to go to bed for a couple of hours and then I’ll be alright for the evening, because I know they advise you not to go to bed don’t they, but I can’t physically not and I find it makes me feel better actually if I do, so for me it works better, so you I’ve learnt to do what suits me rather than what I’m told to do you know they say you muck up your body clock up if you sleep in the day but for me it doesn’t work that way.”*


## 4. Discussion

The objective of this research was to metasynthesize the available qualitative studies exploring how people diagnosed with FM experience and manage poor sleep quality. The overall methodological quality of the included studies was good according to the CASP qualitative checklist.

In this metasynthesis, the authors took the SMT as a conceptual framework and the two overarching themes were pre-established in accordance with two of the domains of the theory “symptom experience” and “symptom management strategies”: (1) experience of poor sleep quality in FM and (2) poor sleep quality management strategies in FM. The sub-themes were also organized taking into account the concepts nested within each of the domains. The results related to the experience of poor sleep quality were classified into two sub-themes: (1) evaluation of poor sleep quality and (2) response to poor sleep quality. The results related to poor sleep quality management strategies in FM were classified into two sub-themes: (1) management strategies to favor sleep and (2) managing the consequences of a sleepless night.

The present analysis showed that poor sleep quality is perceived as a severe symptom of FM and that the most troublesome problems are those related to the maintenance of sleep throughout the night, indicating the experience of fragmented sleep. Likewise, complaints about superficial and non-restorative sleep were common among the participants. When people diagnosed with FM were asked about the meaning of good sleep quality, they usually reported that it was related to having uninterrupted sleep, the feeling of “disconnection” during the night, and waking up feeling refreshed. Therefore, our results showed that for people diagnosed with FM, the most important aspects of sleep that contribute to defining good sleep quality are those related to sleep continuity, the perception of disengagement from the environment, and waking up feeling refreshed in the morning. According to a panel of experts assembled by the National Sleep Foundation [58], the concept of sleep quality is mainly related to sleep continuity, sleep maintenance, sleep initiation, and sleep efficiency. However, because of the subjectivity linked to sleep quality and that its meaning can vary among individuals [59], it is important to continue to explore the meaning that people with different health conditions attach to sleep quality. Moldofsky et al. [60] investigated the meaning of sleep quality among people suffering from insomnia in comparison with good sleepers and concluded that, in general terms, the meaning of sleep quality is very similar between them. However, the lack of quantitatively and qualitatively oriented research comparing the meaning of sleep quality between healthy sleepers, people with other sleep disorders, and people suffering from FM highlights the need to further investigate this subject to provide the most appropriate assessment and management approaches.

Although the beliefs regarding the cause of poor sleep quality have been hardly explored in the available qualitative literature, the results indicated that people diagnosed with FM appear to believe that poor sleep quality development depends on external factors given that most of the participants rely on aspects related to working conditions. 

Another aspect that highlights the severity of poor sleep quality in the context of FM is the perceived effect on other symptoms commonly associated with FM such as pain, fatigue, poor functionality, mood state alterations, and cognitive problems. Particularly, it seems that the most complex relationship for people diagnosed with FM is that of poor sleep quality with fatigue and pain. These beliefs are supported by scientific evidence demonstrating that poor sleep quality correlates with reports of increased fatigue [29]. Likewise, participants argued that fatigue is not alleviated after having a satisfactory quantity of sleep, which may be indicative that the sleep–fatigue relationship could be mediated by other sleep aspects rather than by the total amount of sleep. 

The sleep–poor functionality relationship is perceived as having a negative impact on managing work-related demands, which ultimately leads to a feeling of frustration and failure. The qualitative literature exploring how poor sleep quality affects work performance in people diagnosed with FM is insufficient. However, Litwiller et al. [61] suggested that the consequences of poor sleep quality on affect and cognition mediate the impact of poor sleep quality on task and contextual performance and impair the ability to develop safety behaviors. The authors also highlighted the relevance of including a measure of sleep quality in surveys of wellbeing and satisfaction at the workplace. This information would facilitate the implementation of poor sleep quality management programs that could help employees to develop effective coping strategies.

Regarding the sleep–pain cluster, the results of this metasynthesis suggest that the interaction of these two symptoms is perceived as bidirectional. In this regard, there is an increasing amount of scientific evidence showing that there is a bidirectional relationship between pain and sleep [28,57,62,63,64,65,66]. The proposed underlying physiological mechanisms of the sleep–chronic pain relationship involve multiple brain structures and systems such as the monoaminergic system [65,67], opioid and endocannabinoid systems, the orexinergic system, the immune system, the hypothalamus-pituitary-adrenal axis, and the pineal melatonin system, among others [65].

Other important factors mediating the sleep–pain relationship are psychological processes. Results of a review [62] showed that the cognitive factors associated with pain may be key in the sleep–pain relationship. In 2012, Buenaver et al. [68] investigated whether pain catastrophizing was linked to greater severity of pain and if this relationship could be mediated by sleep problems, including a sample of 214 participants with temporomandibular myofascial disorder. The results showed that pain catastrophizing was, directly and indirectly, related to greater severity of pain. Sleep problems can mediate the indirect relationship between pain catastrophizing and pain intensity, although the mechanisms of this association have not yet been elucidated. Smith et al. [69] postulated that this relationship could derive from the development of ruminative and catastrophic thoughts related to pain just at bedtime. In their research, they found that people with chronic pain think more often about their pain in the moments prior to going to sleep and that a higher incidence of such thoughts can affect the onset and maintenance of sleep. Likewise, people with chronic pain develop negative pre-sleep cognitions related to sleep, although in this case, the authors did not find them to be correlated with problems in sleep continuity.

Concerning the latter, our results showed that, in some cases, people diagnosed with FM respond to poor sleep quality by developing ruminating thoughts about sleep, generating fear of going to bed. Other notable responses are related to the development of negative thoughts as well as feelings of frustration and hopelessness. A repetitive thought is a well-known perpetuating factor in patients with insomnia and Galbiati et al. [70] investigated whether worry and rumination were related to subjective and polysomnographic indices of sleep disruption, comparing a sample of patients with insomnia and a sample of healthy subjects. The results suggested that repetitive thought affected the subjective sleep quality in both samples. However, the results from polysomnographic measures indicated that worry and rumination affected sleep differently in the sample of patients with insomnia. Worry was significantly correlated with problems maintaining sleep, lower sleep efficiency, and decreased total sleep time, while rumination was correlated with an augmented sleep-onset latency and lower sleep efficiency. It seems that perceived stress and dysfunctional beliefs and attitudes about sleep are factors that also affect subjective sleep quality in patients with FM in comparison with healthy people according to the results of Theadom and Cropley [71]. Therefore, exploring in depth the cognitive processes associated with sleep in the context of FM could be key for developing educational materials that respond to the actual needs of these patients and help them to change maladaptive sleep cognitions and to develop effective symptom management strategies.

Regarding poor sleep quality management strategies, our results highlighted that people diagnosed with FM commonly reject pharmacological treatments for both improving sleep quality and dealing with the daytime consequences of poor sleep quality. However, there were also participants reporting drug dependency as a means of falling asleep. The most common worries regarding pharmacological approaches are fear of addiction and side effects. Scientific evidence demonstrated that most of the drugs prescribed in chronic pain patients with concomitant insomnia have several side effects, risk of addiction, cause premature drug withdrawal, and even impact negatively on sleep measures [28]. In this regard, Haack et al. [65] recently investigated the clinical implications of pharmacological and non-pharmacological approaches for the management of the chronic pain–poor sleep quality cluster, highlighting the importance of being cautious in the prescription of sleep-disturbing medications for the treatment of chronic pain. They propose that clinicians must consider the timing of pain medication intake to reduce their detrimental effects on sleep and emphasize the importance of promoting good sleep quality in chronic pain patients through biopsychosocial interventions.

Our results also showed that, although it seems that some people diagnosed with FM have knowledge about the basic principles of sleep hygiene and resort to them to improve their sleep quality, the vast majority tend to develop maladaptive management strategies. The latter may indicate that, at least in some cases, these patients do not receive enough information from health professionals that allows them to develop effective strategies for managing poor sleep quality.

Another important aspect to be investigated in future research involving people diagnosed with FM is whether the beliefs and cognitive responses related to sleep quality affect how they manage poor sleep quality. Research including a sample of people suffering from medically unexplained symptoms, as is the case in FM, found that those people reporting greater understanding of the illness and perceiving that their symptoms are controllable were more prone to developing effective management strategies. Moreover, the development of effective management strategies was correlated with less disability. However, participants showing threat beliefs associated with the illness presented a decrease in active coping responses and were more dependent on external support. Furthermore, these two aspects were significantly correlated with worse disability [72].

Consequently, educational interventions focused on changing the beliefs and cognitive responses of people diagnosed with FM regarding poor sleep quality may be essential to help them to develop effective management strategies. However, our results showed that further qualitative research comprehensively exploring how patients with FM experience and manage poor sleep quality is needed to conceptualize and contextualize sleep education interventions that respond to their needs. 

Engaging in physical exercise practice could also provide an effective strategy not only for improving sleep but also in motivating patients with FM to change maladaptive lifestyle habits. In this regard, a recently published systematic review with meta-analysis [73] analyzed the effects of exercise on fatigue and sleep in FM and which type of exercise is more effective. The results suggested that including meditative exercise programs in the management of FM could provide beneficial effects on sleep quality.

### 4.1. Rigour

In accordance with Sandelowski and Barroso [34] and to fulfill the fifth step of a qualitative metasynthesis on “Maintaining quality control”, we used established methods for the methodological quality assessment (CASP approach) and the analysis of the findings (metasynthesis). The authors also employed a pre-planned and comprehensive search utilizing electronic searches and scanning the list of references of the included studies. Additionally, a peer-review procedure was implemented in the processes of study screening and data extraction. 

To enhance the rigor of the data analysis process, two authors (CCS and GMA) highlighted the participants’ quotations in the included reports that were relevant for responding to the objective of this metasynthesis. Consequently, a meeting with the rest of the research team was organized to reach a consensus regarding the codes and themes that emerged, and the final transcript was modified accordingly.

### 4.2. Limitations of the Study

This qualitative metasynthesis presents some limitations that could have affected the results obtained. The authors did not search grey literature and only included qualitative studies published in English or Spanish, which could have precluded the inclusion of relevant research responding to the objective of this metasynthesis.

## 5. Conclusions

Poor sleep quality is perceived as a severe symptom of FM, with a profound impact on multiple symptoms associated with this chronic health condition such as pain, fatigue, mood, and cognitive complaints, poor functionality, and work performance. The results of this metasynthesis also highlighted that, in most cases, people diagnosed with FM develop maladaptive and ineffective strategies to manage poor sleep quality and its consequences on their general health status. Because of the latter, healthcare professionals should reflect upon whether they are providing enough guidance to these patients that enables them to cope with poor sleep quality more effectively. More effort should be devoted to developing biopsychosocial interventions to alleviate these symptoms. To integrate sleep education interventions in the context of FM, it is imperative to develop future studies that deeply explore the experience of poor sleep quality in people suffering from this health condition.

## Figures and Tables

**Figure 1 jcm-09-04000-f001:**
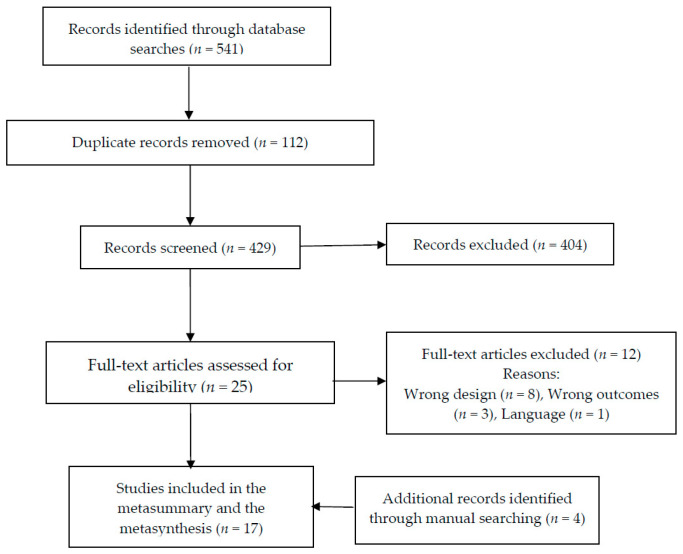
Flow diagram for the identification, screening, eligibility, and inclusion of studies.

**Table 1 jcm-09-04000-t001:** Studies’ characteristics.

Authors and Year of Publication	Country	Sample Characteristics	Method of Approach	CASP Checklist n. of Items Fulfilled/n. of Items
Ramlee et al. [40]	England	*n* = 6 (3♀/3♂) Mean age = 49 years	Personal semi-structured interviews	10/10
Russell et al. [41]	North Ireland	*n* = 14 (12♀/2♂)	Focus groups	10/10
Vincent et al. [42]	USA	*n* = 44 (34♀/10♂) Mean age: 45 years	Open-ended interview administered electronically	10/10
Kleinman et al. [43]		*n* = 34 (30♀/4♂) Mean age = 47.8 years	Focus groups	9/10
Traska et al. [44]	USA	*n* = 8♀ Mean age = 61 years	Interview group	9/10
Sallinen et al. [45]	Finland	*n* = 20♀ Mean age = 54 years	Narrative interview	9/10
Humphrey et al. [46]	USA Germany France	*n* = 40 Mean age = 48.7 years	Open-ended interviews	10/10
Theadom et al. [17]	England	*n* = 16 (14♀/2♂) Mean age (50.95 years)	Semi-structured interviews	9/10
Martin et al. [47]	USA	*n* = 20 (16♀/4♂) Mean age = 50.3 years	Personal structured interviews	9/10
Lempp et al. [48]	England	*n* = 12 (11♀/1♂) Mean age = 49 years	Personal semi-structured interviews	9/10
Arnold et al. [49]	USA	*n* = 48♀ Mean age = 51 years	Focus groups	9/10
Crooks [50]	Canada	*n* = 55♀	Personal semi-structured interviews	9/10
Cunningham et al. [51]	Canada	*n* = 8 (7♀/1♂) Age range 30-70	Personal in-depth interviews	9/10
Söderberg et al. [52]	Sweden	*n* = 25♀ Mean age = 46.8 years	Personal narrative interviews	9/10
Cudney et al. [53]	USA	*n* = 10♀	Unstructured, online support group	9/10
Sturge-Jacobs [54]	Canada	*n* = 9♀ Age range 20-57 years	Personal unstructured interviews	10/10
Raymond and Brown [55]	Canada	*n* = 7 (6♀/1♂)	Personal in-depth semi-structured interviews	9/10

CASP: Critical Appraisal Skills Programme; ♂: male; ♀ female.

**Table 2 jcm-09-04000-t002:** Intrastudy intensity effect sizes and interstudy frequency effect sizes of themes.

Overarching Theme	Experience of Poor Sleep Quality in FM		Poor Sleep Quality Management Strategies in FM		Intrastudy Intensity Effect Sizes
Sub-Themes	Evaluation of Poor Sleep Quality	Response to Poor Sleep Quality	Management Strategies to Favor Sleep	Managing the Consequences of a Sleepless Night	Individual Studies’ Contributions to Sub-Themes
Ramlee et al. [40]	●				0.7%
Russell et al. [41]		●			0.1%
Vincent et al. [42]	●				0.1%
Kleinman et al. [43]	●	●			10.5%
Traska et al. [44]				●	0.1%
Sallinen et al. [45]	●		●		0.2%
Humphrey et al. [46]	●				0.3%
Theadom et al. [17]	●	●	●	●	29%
Martin et al. [47])	●				0.1%
Lempp et al. [48]	●		●		0.2%
Arnold et al. [49]			●	●	0.2%
Crooks [50]	●			●	0.2%
Cunningham et al. [51]	●			●	0.3%
Söderberg et al. [52]	●			●	0.3%
Cudney et al. [53]	●	●	●	●	16.3%
Sturge-Jacobs [54]	●	●	●		0.3%
Raymond and Brown [55]			●		0.1%
**Interstudy Frequency Effect Sizes**					
**Representation of sub-themes in individual studies**	76%	29%	41.6%	41%	

FM: Fibromyalgia; ●: indicates that the sub-theme appeared in the study.

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
