# Peer review of "Poor Sleep Quality Experience and Self-Management Strategies in Fibromyalgia: A Qualitative Metasynthesis"

_jcm, 2020, doi:10.3390/jcm9124000_

Round 1

Reviewer 1 Report

This interesting article describes a metasynthesis of qualitative studies with samples of persons with FM that included content r/t sleep and sleep management. The authors carefully delineated the problem under study, using appropriate methods in their procedures, and discussed the results in an organized manner. 

I would recommend a sentence or two under section 3.1 about the samples obtained (where from, men/women) in order to enhance transferability of findings. 

The discussion had a few disjointed paragraphs that seemed to hone in on working and sleep (not discussed in the background) and drew conclusions which seemed beyond the findings noted. Also, under conclusions, the sentence that contained reference to financial concerns seemed to come out of nowhere. The authors are encouraged to reread all discussion about working conditions and sleep and rework (LL 504-522).

I believe this paper offers new insights into living with FM and suggests potential ideas for management strategies.

Author Response

Dear Reviewer 1:

We are very grateful for your excellent suggestions and comments. We have carefully considered them all and made revisions accordingly. We have addressed each of the reviewer´s requirements as outlined below and are indicated in the manuscript in green color.

This interesting article describes a metasynthesis of qualitative studies with samples of persons with FM that included content r/t sleep and sleep management. The authors carefully delineated the problem under study, using appropriate methods in their procedures, and discussed the results in an organized manner. 

  1. I would recommend a sentence or two under section 3.1 about the samples obtained (where from, men/women) in order to enhance transferability of findings. 

Thank you very much for your observation. The requested information is already presented in Table 1. For this reason, the authors decided not to repeat information that is already available in said table.

  1. The discussion had a few disjointed paragraphs that seemed to hone in on working and sleep (not discussed in the background) and drew conclusions which seemed beyond the findings noted. Also, under conclusions, the sentence that contained reference to financial concerns seemed to come out of nowhere. The authors are encouraged to reread all discussion about working conditions and sleep and rework (LL 504-522).

Thank you for your comment. The authors agree with the reviewer that the relationship between sleep and working performances is no introduced in the background section. This is why the authors added a few lines about this topic in the background section. Please, find it below or in the manuscript (LL 55-58):

A previous qualitative study showed that people diagnosed with FM perceived that poor sleep quality has a great impact on work performance because of their increased need for diurnal rest. Besides, the increased need for diurnal rest also prevents these patients from engaging in social activities [25].

We believe that this topic is well described in the results section and justifies its inclusion in the discussion. However, after revising the discussion, we also believed that some changes were necessary. Please, find the changes below or in the manuscript (LL 415-423):

The sleep-poor functionality relationship is perceived as having a negative impact on managing work-related demands which ultimately leads to a feeling of frustration and failure. The qualitative literature exploring how poor sleep quality affects work performance in people diagnosed with FM is insufficient. However, Litwiller et al. [60] suggested that the consequences of poor sleep quality on affect and cognition mediate the impact of poor sleep quality on task and contextual performance and impair the ability to develop safety behaviors. The authors also highlighted the relevance of including a measure of sleep quality in surveys of well-being and satisfaction at the workplace. This information would facilitate the implementation of poor sleep quality management programs that could help employees develop effective coping strategies.

The authors also rewrote the conclusions section according to the recommendation of the reviewer. Please, find it bellow or in the manuscript (LL 518-524):

Poor sleep quality is perceived as a severe symptom of FM causing a profound impact in multiple symptoms associated with this chronic health condition such as pain, fatigue, mood, and cognitive complaints, poor functionality, and work performance. The results of this meta-synthesis also highlighted that in most cases people diagnosed with FM develop desperate and ineffective strategies to manage poor sleep quality and its consequences on their general health status. Because of the latter, healthcare professionals should reflect if they are providing enough guidance to these patients that facilitates them coping with poor sleep quality more effectively. […]

I believe this paper offers new insights into living with FM and suggests potential ideas for management strategies.

In the name of the authors, I hope that this revised manuscript answers all the concerns contained in the reviews, and we are grateful for the thought and effort the reviewer has put into these reviews.  

Yours sincerely,

Filip Bellon

Reviewer 2 Report

Thank you for the opportunity to review this manuscript. Sleep is an under explored areas in all chronic diseases, and in particular as it related to those with lived experience. Improved clinician understanding of these experiences and the strategies tried which were both successful and unsuccessful. I have organized my feedback into large headings of the manuscript.

Generally, the initial sections of the article are well written, yet some spelling and grammar errors are present. I suggest a careful re-read prior to re-submission with careful attention given to missing letters, pronouns, length of sentences, and that paragraphs are more than one sentence to improve readability.

The introduction presents most of the information required to make the case for the metasynthesis and uses appropriate referencing. I would however suggest more information be provided as to why a metasynthesis was chosen for this review and the purpose of metasyntheses to be added.

I would also suggest removing the paragraph explaining the data analysis process and framework, and adding it to the methods section. Further details about the framework and why it was chosen would be helpful. Many biopsychosocial conceptual frameworks exist in the health care world. So why this one for this metasynthesis?

Methods: 

Research question: The question should be further integrated into the sentence to improve readability. 

Inclusion criteria: Were children included? Was an age range targeted? Should am inclusion criterion be adults?

Data source: Could you please elaborate on how the manual search was conducted and when?

Analysing and synthesizing findings: A definition with quotation marks  is provided for qualitative metasummary process, yet a page number is not included. 

Results: The PRISM flow chart is well presented. However, a short summary of the characteristics of the studies  (e.g., total number of patients, type of study design used, countries where the studies were conducted) would be appreciated. This section is complex, and I commend the authors on their organization of it. 

Metasynthesis: Although the 2 major themes - experience of poor sleep quality in FM and Sleep management strategies in FM  have been highlighted, they have not been defined anywhere in this section. The sub-themes for each theme also do not have clear definitions. Furthermore , each sub-theme  have multiple sub-sub-themes, many which over lap with each other. Although I appreciate that this is quite arduous work, I believe that more analysis is required to move further beyond the patterns identified and deconstructing the individual findings and provide new meaning to the experience of experience and management strategies for poor sleep quality used by people diagnosed with FM. More specifically, I do think that some of the sub-sub-themes could be condensed into each other to provide richer definitions and accounts of each theme. Furthermore, I would also encourage authors in the text to select only one quote that is representative of the idea for each them. Other quotes are already contained in the supplementary file, which you have made already made available to the reader. When quotes are used and if using italics to highlight them, I encourage the authors to be consistent with this strategy throughout this section.

Finally, I would also caution the authors on their use of quantitative terminology throughout this manuscript. Moreover, words such as mediate or modulate are not terminology typically seen in qualitative study designs, or data analysis. I would suggest this type of terminology be revised and substituted with other words. 

The following references may provide the authors with a model upon which to base the suggested revisions.

Snelgrove, S., & Liossi, C. (2013). Living with chronic low back pain: a metasynthesis of qualitative research. Chronic illness, 9(4), 283-301;

Crowe, M., Whitehead, L., Seaton, P., Jordan, J., Mccall, C., Maskill, V., & Trip, H. (2017). Qualitative meta‐synthesis: the experience of chronic pain across conditions. Journal of Advanced Nursing, 73(5), 1004-1016;

Jordan, A., Family, H., & Forgeron, P. (2017). Interpersonal relationships in adolescent chronic pain: A qualitative synthesis. Clinical Practice in Pediatric Psychology, 5(4), 303.

Discussion: Much of the discussion centrered around the mechanisms associated with poor sleep quality. Some references were made to the strategies employed by people with FM to address poor sleep quality. According to the literature, is there a role for co-designing this program with patients? 

I thank the authors for embarking on this much needed, yet complex literature synthesis. Without it, a clear path for future research would be difficult to identify and strategies to further alleviate the severe impacts of poor sleep in people diagnosed with FM would remain out of reach.

Thank you for the opportunity to revise this manuscript. 

Author Response

Dear Reviewer: 

We are very grateful for the excellent suggestions and comments from the reviewers. We have carefully considered them all and made revisions accordingly. We have addressed each of the reviewer's requirements as outlined below and are indicated in the manuscript in green color.

Thank you for the opportunity to review this manuscript. Sleep is an under explored areas in all chronic diseases, and in particular as it related to those with lived experience. Improved clinician understanding of these experiences and the strategies tried which were both successful and unsuccessful. I have organized my feedback into large headings of the manuscript.

  1. Generally, the initial sections of the article are well written, yet some spelling and grammar errors are present. I suggest a careful re-read prior to re-submission with careful attention given to missing letters, pronouns, length of sentences, and that paragraphs are more than one sentence to improve readability.

The authors really appreciate the comprehensive reading of the document by the reviewer. As suggested, we have revised the spelling and grammar errors of the document and made the appropriate corrections. Besides, the authors also rewrote some sentences to reduce their length. Please, find all the changes highlighted in green color throughout the document.

  1. The introduction presents most of the information required to make the case for the metasynthesis and uses appropriate referencing. I would however suggest more information be provided as to why a metasynthesis was chosen for this review and the purpose of metasyntheses to be added.

Thank you for your observation. In accordance with your suggestion, the authors have added a few lines justifying why we adopted a metasynthesis. Please, find it bellow or in the document in green color (LL 79-89):

Qualitative studies are rarely aimed at having a direct impact on healthcare practice or policymaking. However, conducting systematized reviews of qualitative studies could be a valuable method in facilitating the transferability of qualitative data to improve healthcare attention [30]. Although the vast majority of meta-syntheses of qualitative studies are published in the fields of nursing and sociology, there is a clear trend towards the publication of metasynthesis in all disciplines of health science [31].

Therefore, the authors considered it appropriate to carry out a metasynthesis aimed at summarizing the available qualitative research exploring the experience and management of poor sleep quality in people diagnosed with FM. Carrying out this meta-synthesis could improve the understanding of the phenomenon of poor sleep quality in the context of FM and provide valuable information for the development of treatment strategies. 

  1. I would also suggest removing the paragraph explaining the data analysis process and framework, and adding it to the methods section. Further details about the framework and why it was chosen would be helpful. Many biopsychosocial conceptual frameworks exist in the health care world. So why this one for this metasynthesis?

Thank you for your comment. The authors have added the paragraph explaining the data analysis process and framework to the methods section. Besides, we also have provided further details about the framework and why it was chosen in this metasynthesis. Please, find the changes below or in the document highlighted in green color (LL 149-160):

In this metasynthesis we took the Symptom Management Theory (SMT) [37] as a biopsychosocial conceptual framework. The SMT is a deductive, middle-range theory developed by Larson et al. [38] providing a new model for nurses to assess the patients’ symptom experience in a comprehensive and multidimensional manner. Besides, the SMT is also aimed at providing a framework for the development of symptom management strategies and to evaluate their outcomes in the biological, psychological, and social spheres of the patient.

As previously stated, FM is a complex chronic health condition for which curative treatments are currently nonexistent. Therefore, the main approach is focused on developing symptom management strategies that help alleviate the severity of symptoms such as pain, poor sleep quality, fatigue, and mood disturbances. Because of the latter, a symptom-focused conceptual framework could help to understand how people diagnosed with FM experience the symptoms that characterize this health condition.

Methods:

  1. Research question: The question should be further integrated into the sentence to improve readability.

Thank you for your suggestion. The authors have rewritten this section as it follows and it is also available in the document (LL 107-109):

There is a necessity for increasing the understanding of the phenomenon of poor sleep quality in the context of FM. Therefore, this metasynthesis was developed to answer the research question “How people diagnosed with fibromyalgia experience and manage poor sleep quality?”.

  1. Inclusion criteria: Were children included? Was an age range targeted? Should am inclusion criterion be adults?

Thank you for your suggestion. The authors already stated in the section that the samples of the studies must be adults in line 116 of the document. However, for further clarification, we added a short statement in criterion number 2 about this. Please, see it below or in the document in green color (line 117-118):

Studies totally o partially exploring the experience and/or management related to poor sleep quality in adult people diagnosed with FM

  1. Data source: Could you please elaborate on how the manual search was conducted and when?

Thank you for your comment. We appreciate that the reviewer found this mistake because the authors did not perform a hand-search but they only scanned the reference list of the included studies. Please, find the correction below or in the document in green color (LL 122-123):

The reference lists of the included studies were scanned to identify additional studies.

  1. Analysing and synthesizing findings: A definition with quotation marks is provided for qualitative metasummary process, yet a page number is not included.

The authors believe that the page numbers for the quotation included in this section are available in the reference:

Sandelowski, M.; Barroso, J. Synthesizing Qualitative Research Findings. In Handbook for synthesizing qualitative research; Springer Publishing Company, LLC: New York, 2007; pp. 151–226 ISBN 0826156940.

Results:

  1. The PRISM flow chart is well presented. However, a short summary of the characteristics of the studies (e.g., total number of patients, type of study design used, countries where the studies were conducted) would be appreciated. This section is complex, and I commend the authors on their organization of it.

Thank you very much for your observation. The requested information is already presented in Table 1. For this reason, the authors decided not to repeat information that is already available in said table.

  1. Metasynthesis: Although the 2 major themes - experience of poor sleep quality in FM and Sleep management strategies in FM have been highlighted, they have not been defined anywhere in this section. The sub-themes for each theme also do not have clear definitions. Furthermore , each sub-theme  have multiple sub-sub-themes, many which over lap with each other. Although I appreciate that this is quite arduous work, I believe that more analysis is required to move further beyond the patterns identified and deconstructing the individual findings and provide new meaning to the experience of experience and management strategies for poor sleep quality used by people diagnosed with FM. More specifically, I do think that some of the sub-sub-themes could be condensed into each other to provide richer definitions and accounts of each theme. Furthermore, I would also encourage authors in the text to select only one quote that is representative of the idea for each them. Other quotes are already contained in the supplementary file, which you have made already made available to the reader. When quotes are used and if using italics to highlight them, I encourage the authors to be consistent with this strategy throughout this section.

The effort that the reviewer put into this section is very appreciated. Your comments and suggestions helped us to reconsider our analysis and presentation of the results. In accordance with your suggestions, we have reorganized and rewritten all sub-sub-themes. As we made amendments throughout the whole section, here we summarize the changes that we made and invite the reviewer to check them in the manuscript.

  • The authors provided a definition for all the themes and subthemes
  • We reanalyzed the findings and rewritten some sections in the hopes that they provide a description of the experience and management of poor sleep quality that goes beyond individual findings.
  • We also condensed some sub-sub-themes into each other as it follows:
    1. In the sub-theme “response to poor sleep quality” we have condensed the sub-sub-theme “in despair” into the sub-sub-theme “feeling frustrated and a failure”. (LL 287-301). We have also condensed the sub-sub-theme “Ruminating thoughts” into the sub-sub-theme “Fear of going to bed”. (LL 302-312)
    2. Originally, there were three sub-themes included in the section “Poor sleep quality management strategies in FM”. However, we decided to reduce them to two sub-themes and were renamed as 1) Management strategies to favor sleep, and 2) Managing the consequences of a sleepless night. In this way, the sub-themes “strategies to fall asleep” and “nocturnal strategies” are now integrated together into the sub-theme “Management strategies to favor sleep”. (LL 320-351)
    3. Additionally, the sub-sub-themes “establishing regular sleep schedules” and “looking for answers” are now integrated into a sub-sub-theme named “Self-management: behavioral adaptations” (LL 333-351)
  • The authors selected only one quote that is representative of the idea for each theme/sub-theme
  • We changed the format style of the quotations and they are now all in italics

  1. Finally, I would also caution the authors on their use of quantitative terminology throughout this manuscript. Moreover, words such as mediate or modulate are not terminology typically seen in qualitative study designs, or data analysis. I would suggest this type of terminology be revised and substituted with other words. The following references may provide the authors with a model upon which to base the suggested revisions.

Snelgrove, S., & Liossi, C. (2013). Living with chronic low back pain: a metasynthesis of qualitative research. Chronic illness, 9(4), 283-301;

Crowe, M., Whitehead, L., Seaton, P., Jordan, J., Mccall, C., Maskill, V., & Trip, H. (2017). Qualitative meta‐synthesis: the experience of chronic pain across conditions. Journal of Advanced Nursing, 73(5), 1004-1016;

Jordan, A., Family, H., & Forgeron, P. (2017). Interpersonal relationships in adolescent chronic pain: A qualitative synthesis. Clinical Practice in Pediatric Psychology, 5(4), 303.

Thank you for your suggestion and for providing the references. We have modified the quantitative terminology throughout the manuscript.

Discussion:

  1. Much of the discussion centrered around the mechanisms associated with poor sleep quality. Some references were made to the strategies employed by people with FM to address poor sleep quality. According to the literature, is there a role for co-designing this program with patients?

We would like to apologize because the authors did not understand this comment.

I thank the authors for embarking on this much needed, yet complex literature synthesis. Without it, a clear path for future research would be difficult to identify and strategies to further alleviate the severe impacts of poor sleep in people diagnosed with FM would remain out of reach.

Thank you for the opportunity to revise this manuscript.

In the name of the authors, I hope that this revised manuscript answers all the concerns contained in the reviews, and we are grateful for the thought and effort the reviewer has put into these reviews.  

Yours sincerely,

Filip Bellon

Reviewer 3 Report

This study employed principles of metasynthesis to evaluate how fibromyalgia patients  experience and manage poor sleep quality. The conclusion is that poor sleep is a disabling symptom in FM, negatively affecting the patients’ general health status, with commonly employed treatments being perceived as generally non effective.

The study is interesting and original, as it focuses on a relatively neglected aspect of FM symptoms. I have the following comments :

-Physical exercise is regarded as an important therapeutic aspect in FM. It has been shown to be moderately effective in lowering fatigue and also in providing some effect of enhancement of sleep quality in fibromyalgia (see Effectiveness of Exercise on Fatigue and Sleep Quality in Fibromyalgia: A Systematic Review and Meta-analysis of Randomized Trials Arch Phys Med Rehabil  2020 Jul 25;S0003-9993(20)30434-2  doi: 10.1016/j.apmr.2020.06.019. Online ahead of print ). Physical exercise also influences beta-endorphin levels,  and in turn beta-endophin levels have been shown to be impaired in fibromyalgia (see Peripheral blood mononuclear cell β-endorphin concentration is decreased in chronic fatigue syndrome and fibromyalgia but not in depression: Preliminary report. Clinical Journal of Pain.Volume 18, Issue 4, 2002, Pages 270-273).

In the discussion I think it would be opportune for the authors to comment on the important relationship between impaired sleep, physical exercise and beta endorphin levels in fibromyalgia,  with relavant refs quoted.

-The English language contains some mistakes/imperfections, that should be corrected. Just a few examples:

-Line 42 ….given the evidences of a hyperexitable Central Nervous System… It should be…given the evidence of a hyperexcitable Central Nervous System..

-Line 65… Poor sleep show a strong tendency.. It should be: Poor sleep shows…

-Line 485 : …usually report that is related to having… it should be : that it is related to having…

-Line 197… although it seems that is not a problem… It should be:… that it is not a problem…

-Line 504 : Another aspect that highlight the severity of poor sleep… it should be :.. that highlights….

Author Response

Dear reviewer:

We are very grateful for your excellent suggestions and comments. We have carefully considered them all and made revisions accordingly. We have addressed each of the reviewer's requirements as outlined below and are indicated in the manuscript in green color.

This study employed principles of metasynthesis to evaluate how fibromyalgia patients experience and manage poor sleep quality. The conclusion is that poor sleep is a disabling symptom in FM, negatively affecting the patients’ general health status, with commonly employed treatments being perceived as generally non effective.

The study is interesting and original, as it focuses on a relatively neglected aspect of FM symptoms. I have the following comments:

  1. Physical exercise is regarded as an important therapeutic aspect in FM. It has been shown to be moderately effective in lowering fatigue and also in providing some effect of enhancement of sleep quality in fibromyalgia (see Effectiveness of Exercise on Fatigue and Sleep Quality in Fibromyalgia: A Systematic Review and Meta-analysis of Randomized Trials Arch Phys Med Rehabil 2020 Jul 25; S0003-9993(20)30434-2 doi: 10.1016/j.apmr.2020.06.019. Online ahead of print). Physical exercise also influences beta-endorphin levels, and in turn beta-endorphin levels have been shown to be impaired in fibromyalgia (see Peripheral blood mononuclear cell β-endorphin concentration is decreased in chronic fatigue syndrome and fibromyalgia but not in depression: Preliminary report. Clinical Journal of Pain. Volume 18, Issue 4, 2002, Pages 270-273).

In the discussion I think it would be opportune for the authors to comment on the important relationship between impaired sleep, physical exercise and beta endorphin levels in fibromyalgia, with relavant refs quoted.

Thank you for your suggestion. The authors included the first reference that the reviewer recommended and included a few lines about physical exercise as a potential therapeutic approach in improving sleep quality. Please, find it below or in the manuscript (LL 494-499):

Engaging in physical exercise practice could also provide an effective strategy not only for improving sleep but also in motivating patients with FM to change maladaptive lifestyle habits. In this regard, a recently published systematic review with meta-analysis [73] analyzed the effects of exercise on fatigue and sleep in FM and which type of exercise is more effective. The results suggested that including meditative exercise programs in the management of FM could provide beneficial effects on sleep quality.

However, we decided not to discuss the relationship of physical exercise and endorphin levels as we believe it is a controversial topic. Taking into account a more recently published study, it was shown that the increase in the levels of endorphins in FM patients post-exercise is not significant in comparison with healthy controls. This is why, we believe that prior to making any statement in this regard, further research supporting the idea is necessary:

Korean J Pain. 2016 Oct; 29(4): 249–254.

Published online 2016 Sep 29. doi: 10.3344/kjp.2016.29.4.249

PMCID: PMC5061641

PMID: 27738503

The acute effect of maximal exercise on plasma beta-endorphin levels in fibromyalgia patients

Ali Bidari,corresponding author Banafsheh Ghavidel-Parsa,* Sahar Rajabi,† Omid Sanaei,‡ and Mehrangiz Toutounchi†

  1. The English language contains some mistakes/imperfections, that should be corrected. Just a few examples:

-Line 42 ….given the evidences of a hyperexitable Central Nervous System… It should be…given the evidence of a hyperexcitable Central Nervous System.

-Line 65… Poor sleep show a strong tendency. It should be: Poor sleep shows…

-Line 485 : …usually report that is related to having… it should be : that it is related to having…

-Line 197… although it seems that is not a problem… It should be:… that it is not a problem…

-Line 504 : Another aspect that highlight the severity of poor sleep… it should be :.. that highlights….

Thank you very much for your observation. The authors have reviewed the whole document to correct the mistakes regarding the English language. Apart from the mistakes highlighted by the reviewer, we also carried out a comprehensive evaluation of the use of English throughout the document and made the proper corrections. The changes are highlighted in green color throughout the whole document.

In the name of the authors, I hope that this revised manuscript answers all the concerns contained in the reviews, and we are grateful for the thought and effort the reviewer has put into these reviews.  

Yours sincerely,

The corresponding author